# Development of Digital Competences in Students of a Public State-Owned Chilean University Considering the Safety Area

**Marcelo Rioseco Pais [1,\*], Juan Silva Quiroz [2] and Claudia Carrasco-Manríquez [3]**

[1] Research Directorate, University of Talca, Talca 3460000, Chile
[2] Faculty of Humanities, Universidad de Santiago de Chile, Santiago 8330015, Chile; juan.silva@usach.cl
[3] Faculty of Education, University of Talca, Talca 3460000, Chile; claudia.carrasco@utalca.cl
[\*] Correspondence: marcelo.rioseco@utalca.cl

**Abstract:** The present study was carried out in a Chilean public university with the purpose of describing digital competences in safety area and their relationship with contextual variables in first-year students of different programs. In this research study, we processed the competences of the safety area of the COMPDIG-PED test, which is based on the European reference framework for digital skills DIGCOMP. The instrument was applied to 4360 higher education students, attending undergraduate courses in 2021 and 2022. The results indicated a relationship between the context variables analyzed through the COMPDIG-PED test and the digital safety competences. This relationship is linked to higher scores in the female gender, in students with more years of experience in the use of digital technologies, in those who came from private paid or subsidized institutions, and in those who were trained in a humanist scientific education.

**Keywords:** digital competence; access to technology; cybersecurity; DIGCOMP; higher education

## 1. Introduction

The International Telecommunication Union (ITU) (2023), the United Nations' specialized agency for information and communication technologies, reported that, in 2021, at least 97% of the world's population had access to 2G technology, 95% to 3G, 88% to LTE/WiMAX, and 19% had access 5G/6G networks. According to the same source, in 2022, 66% of the world's population, equivalent to 5.3 billion people, used the internet. In countries like the US, this percentage was 91%, in Spain it was 94%, and in Germany it was 91%, while in the United Arab Emirates, Qatar, Kuwait, and Iceland, it reached 100%. In Spain, the majority of students possess a smart mobile device that enables them to play audiovisual content, complete tasks, and leverage advanced technologies such as virtual and augmented reality without incurring additional expenses or utilizing supplementary devices [1]. In the case of Chile, 99% have access to 2G, 95% to 3G, and 88% to LTE/WiMAX. Likewise, three-quarters of the world population aged 10 or over in 2022 owned a mobile phone.

Without a doubt, information and communication technologies (ICTs) have experienced rapid growth in recent decades, affecting the lives of people and society. Hand in hand with ICTs, the forms of production, commerce, work, education, knowledge related to health and the body, and the way people interact and establish personal relationships have been changing. The impact that ICTs have today is enormous. However, as a digital ecosystem develops and millions of human beings worldwide share data and information, new requirements and threats related to security also appear [2,3].

For Baca [4], computer security is the discipline that, based on internal and external company policies and regulations, is responsible for protecting the integrity and privacy of the information stored in a computer system against any type of threats, minimizing both the physical and logical risks to which it is exposed. This definition can be extended beyond the company and incorporate other types of organizations, such as government entities or

non-governmental organizations (NGOs), and people who, although not carrying out their activities in the digital world based on policies and standards, implement strategies and incorporate habits related to computer security [5].

Currently, computer security is a growing problem. By way of example, in 2010, fewer than 50 million executable programs with malicious code or "malware" were known. This number doubled in 2012, and, in 2019, it exceeded 900 million (Sarker et al., 2020). Consumers and citizens are regular victims of cybercrime. A recent study conducted by CSIS [6] estimated that cybercrime costs approximately 1% of global GDP, equivalent to USD 600 billion each year. In the United States, according to data from the Federal Bureau of Investigation [7], almost 850,000 complaints were filed, and USD 7 billion were defrauded through cybercrime, with phishing or identity theft of users leading the way, along with their vishing and smishing variants. These figures are triple the number of complaints received in 2017 and multiply their economic impact of just five years ago by five. In Latin America, in 2022, cyberattacks increased significantly in relation to previous years. According to this report, the most affected countries in the region are Peru, with 18% of the attacks, followed by Mexico, with 17%, and Colombia, with 12%, corresponding to 10% of those reported globally. On the other hand, a study by the Inter-American Development Bank [8] on cybersecurity in Latin America and the Caribbean shows that the region is unprepared to handle cyberattacks. A total of 7 of the 32 countries studied have a critical infrastructure protection plan, while 20 have established cybersecurity incident response teams. In Europe, in 2021, the United Kingdom led the density of cybercrimes, with 4783 victims, 40% more than in 2020. In 2020, in Spain, police forces were aware of 287,963 criminal acts related to cybercrime, although the figure may be much higher due to under-reported crimes. This means there was an increase of 31.9% compared to 2019, which was 35.8% higher than 2018, which indicates a worrying evolution [8].

In the case of Chile, the cybercrime brigade of the Investigative Police [9] warned that in 2021 there was an 89% growth in complaints about cybercrimes. According to the PDI, the COVID-19 pandemic impacted the bulk of this figure because, comparing 2019 with 2020, the increase was 29%. In any case, the growth in the number of cases and complaints increases each year in a sustained manner. According to the report "The State of Ransomware 2022", delivered by the cybersecurity company Sophos, 63% of companies in Chile have been affected by cybercriminals in 2021, a notable increase compared to the 33% registered in 2020 [10].

However, cyberattacks and cybercrime are not the only security-related problems that appear along with the spread of ICTs. There are also effects and consequences on people's health, changes in lifestyle and consumption habits, and the environment [10].

According to Terán [11], DTs are generating new addictions related to excessive use. Some people show symptoms shared by any addiction, such as compulsive desire, decreased ability to control consumption, withdrawal syndrome, tolerance, progressive abandonment of other sources of pleasure or entertainment, or persistence in substance use despite its evident harmful consequences. Due to the magnitude of the problem, the WHO included addiction to video games for the first time in its list of mental disorders in the latest International Classification of Diseases, eleventh revision (ICD-11).

In Spain, for example, the ESTUDES 2016–2017 survey, conducted by the National Plan on Drugs in the school population aged 14 to 18 [12], identified compulsive use of the internet in 21% of the students, with a score higher than 28 on the Compulsive Internet Use Scale (CIUS), 4.6% more than the 2014 survey, while gambling on and off the internet affected 6.4% and 13.6%, respectively. Along the same lines, said delegation indicates that between 0.2% and 12.3% of adolescents meet the criteria for problem gambling.

In conclusion, Tucho and González [13] argue that the consumption of technological devices bears significant environmental and socioeconomic consequences, primarily due to the use of conflict minerals, which are extracted under inhumane conditions, and the manufacturing process taking place in impoverished countries that lack basic safety standards. Additionally, the growing number of devices and data centers contributes to

an overall increase in energy consumption, further exacerbating these negative impacts. Another point is that all these devices subsequently become electronic waste, usually dumped illegally in poor regions of the planet.

On the other hand, other researchers believe that the effect of the ICTs on the environment is not negative, such as Walid Chatti [14], who, between 2002 and 2014, explored the connections between ICTs, transportation, and CO2 emissions in 43 countries and concluded that the use of the ICTs in the transportation sector, if well adapted, decreases pollution.

In any case, beyond the conclusions reached when studying the phenomenon of the massification of ICTs, it is increasingly necessary for citizens to know the digital skills required today, particularly the ones related to computer security, understood in a broad sense.

The purpose of this study was to explore and characterize the digital competence of computer safety among first-year undergraduate students in a Chilean public university. This research aims to understand how this competence is associated with various contextual variables, including the year of entry to the university, the gender of the students, the number of years they have had access to digital technologies, and the type of school system in which their mandatory schooling was conducted. The hypothesis of the study is that digital competence in the area of safety is related to the contextual variables of first-year university students.

Evidence generated in this research study aims to provide a deeper understanding of students' digital skills and the factors that influence them. Findings can inform and guide the development of educational policies, curriculum planning, and the implementation of training programs. Ultimately, the findings of this study can contribute to improving the ability of students to face challenges and opportunities in the digital world. Therefore, this study is important in today's digital age, where computer literacy has become an essential component of students' education and development.

*Digital Competences, Safety Area*

To address the issue of security in digital environments, we must have the risks generated by the internet and digital technology in mind without falling into technophobic discourses, but we must take into account the necessary measures when sharing information and take into account the physical and psychological effects that excessive hours on the internet can produce [15].

The big problem with cybersecurity is habitually skipping protection measures. The end user is always the weakest link, especially if their digital competences are not sufficiently developed. For this reason, it is essential to educate people to use digital devices and tools and take protective measures and strategies necessary for this interconnected world, which goes beyond protecting passwords or identity in social networks; it includes aspects such as health care or environmental sustainability.

Digital competence (DC) is a set of capacities encompassing knowledge, skills, attitudes, strategies, and values that allow users to get the most out of digital technologies to perform tasks, solve problems, and communicate effectively [16,17]. Digital competence is not simply limited to the acquisition of technical knowledge or specific skills, but also involves a series of strategies and values that are essential for effective and ethical interaction with digital technology. These strategies are approaches or methods that users employ when using technology effectively and efficiently. Digital competence has been identified as one of the eight critical life skills, along with communication in the first language, communication in foreign languages, mathematical and basic science, technology competences, learning to learn, social and civic competences, sense of initiative, and entrepreneurial spirit [18].

There are various competency certification programs developed and promoted by governmental and non-governmental organizations. The European Computer Driving License (ECDL), also known as International Computer Digital Literacy (ICDL), is a certifi-

cation program for computer knowledge related to the essential ability of a standard user to handle information technology [19].

Likewise, DIGCOMP is a digital competence framework created by the European Commission, which proposes a set of digital competences for all citizens. The DigComp framework creates an agreed vision of which competences people need to overcome the challenges of digitization in almost all aspects of modern life [20]. It aims to create a common language that can be used in tasks ranging from policy formulation and goal setting to planning, evaluation, and monitoring of teaching.

DIGCOMP acknowledges five areas of digital competences: search and management of information and data; communication and collaboration; digital content creation; safety; and problem solving. The first three refer to skills that can be perceived in specific activities and uses. The last two are transversal, because they apply to any type of activity carried out through digital media.

The DIGCOMP framework considers a globalizing concept of DC and is currently used in various investigations to assess DCs at the university level in general [21–24].

The safety area considers four competences (Figure 1): device protection; content, personal data, and privacy in digital environments; physical and psychological health and awareness of the impact of digital technologies on well-being and social inclusion; and finally, awareness of the environmental impact of digital technologies and their use. In turn, each of these areas defines four levels of performance: basic, intermediate, advanced, and highly specialized. They also distinguish knowledge from skills and attitudes and describe the application of the competence in a working and a formative setting.

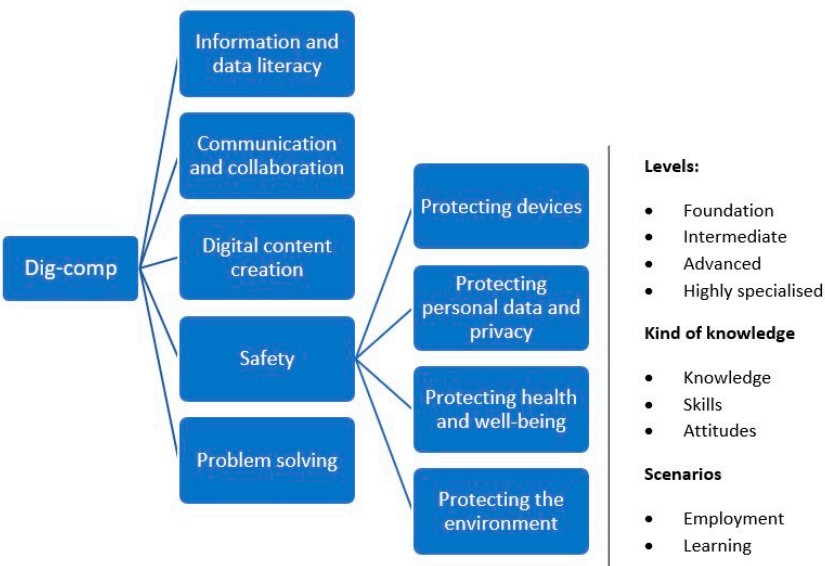

**Figure 1.** Digital competences, safety areas.

In a study with education students from three universities in Chile through a DIGCOMP-based instrument to measure the DC level, Silva and Morales [24] found that the areas with the highest levels of achievement were "network safety" and "online communication and collaboration". On the contrary, the lowest levels of achievement were reached in the "digital information and literacy", "digital content creation", and "problem solving" areas. In a study by González-Calatayud et al. [22] to improve the DCs of second-year pedagogy students through tasks focused on working on each of the DIGCOMP areas, students generally showed an average level of digital competence in all areas. In "problem solving", "information and data literacy", and "digital content creation", the authors found the lowest average and obtained higher values in "communication and collaboration" and "network safety". Rebollo-Catalán, Mayor-Buzon, and García-Pérez [25] researched women's use of social networks. The results showed that most women surveyed believe they have

a medium-high level of digital competences in using social networks. They highlight emotional, functional, and network security competences while presenting deficiencies in informational and creative competences.

## 2. Materials and Methods

The general objective of the research was to describe the level of digital competences of the safety area in first-year undergraduate students of a Chilean public university and their relationship with context variables: students' year of admission to the university, gender, years of access to digital technologies, and type of educational center of origin, from the point of view of dependence and education modality.

Specific objectives:

- Studying the relationship between the level of achievement of the digital competences of safety and the admission years: 2021 and 2022.
- Studying the relationship between the level of achievement of the digital competences of safety and gender.
- Studying the relationship between the level of achievement of the digital competences of safety and the number of years of access to digital technologies.
- Studying the relationship between the level of achievement of the digital competences of safety and students' educational origin.
- Studying the relationship between the level of achievement of the digital competences of safety and students' educational background.

For this research, we chose the first-year students (newcomers) because it is of interest to know their DCs at the beginning of the process, in such a way as to have this information in a timely manner, to develop the DCs throughout university education.

### 2.1. Instrument to Assess Digital Competence in Pedagogy Students

Currently, various instruments can assess DCs in university students, such as INCOTIC 2.0 [26], created in Spain and designed to perform a self-diagnostic assessment of DCs in first-year undergrads [26]. In Latin America, an adapted version is used: INCOTIC-LA [27], which is in the pilot stage, being applied in universities on the continent. Another instrument is ACUTIC [28], developed to study university students' attitudes toward ICTs [29].

Most instruments applied to describe digital competences are of self-perception, in which students have the highest perceptible level. However, self-perception instruments present several limitations that should be considered. First, participants could provide inaccurate or biased responses due to factors such as insincerity, the desire to please, or the influence of social norms. Also, self-perception can change; therefore, participants' responses may fluctuate over time, making it difficult to obtain consistent measures. In addition, self-perception instruments are often based on predefined rating scales or categories, which may limit participants' ability to express their experience. In summary, although self-perception instruments are valuable for exploring subjective phenomena in the social sciences, it is essential to be aware of their limitations and to approach them in a critical and, hopefully, complementary manner with other sources of information. For this reason, COMPDIG-PED was designed as a standard assessment instrument, which, unlike an appreciation scale, assesses the responses to specific situations of DC deployment in dichotomous parameters (correct and incorrect). This instrument was applied to pedagogy course newcomers from Chilean public universities to assess their DC level. Currently, the instrument is being used at the University of Santiago de Chile to measure DCs in first-year grads of various majors. This work is based on the results obtained in the years 2021 and 2022.

The COMPDIG-PED assessment instrument was designed to measure the DCs of university students in the Chilean higher education context. It showed reliability and validity and allowed the reported information to guide institutions in specific curricular improvement plans.

For the construction of the COMPDIG-PED digital competence assessment instrument, DIGCOMP was used as the reference framework, designed to generate a common reference framework regarding the understanding and development of digital competences in Europe [29]. Specifically, we used DIGCOMP 2.1 [20]. Adopting a non-experimental cross-sectional mixed approach, the instrument design safeguarded various pieces of evidence on its validity [30]. This study adopts the DIGCOMP digital competencies framework because of its multidimensional, cross-cutting, and holistic approach. This approach considers not only technical knowledge, but also the cognitive, emotional, and social skills that users need to effectively apply digital technology in various contexts. These characteristics make DIGCOMP a widely used and recognized framework for assessing and developing digital competencies in individuals, organizations, and educational institutions. The method was developed in the following stages: preliminary design of the instrument, application to a representative sample, and empirical analysis of validity evidence.

The test-type assessment instrument was prepared, consisting of closed and multiple choice questions. The test considered specific situations of the use of digital technologies in the personal and academic context, relevant to the local reality. To safeguard the rigor, accuracy, and content validity of the questions prepared, they were submitted to expert judgment [31]. The criteria-based content analysis made it possible to estimate how much specialists agreed on the clarity of the instructions and wording of the items, the lack of need for recall, freedom from bias, and the adequacy of the response categories [32]. Five experts in the scope of higher education participated, four representing Chile and one representing Spain. This process was carried out through validation matrices, where each expert analyzed the validity conditions with a Yes (1) or No (0): pertinence, relevance, and writing. From the scores assigned by the experts, we could establish the overall quality of the question, with variations from 73% to 100%. The three items best evaluated by the experts were left for each competence, all above 80%.

The final instrument comprised 63 items, and the 3 items for each of the 21 competences were distributed into five areas, as follows: area 1: information and data literacy, 9 items, from 1 to 9; area 2: online communication and collaboration, 18 items, from 10 to 27; area 3: digital content creation, 12 items, from 28 to 39; area 4: network safety, 12 items, from 40 to 51; and area 5: problem solving, with 12 items, from 52 to 63.

To determine the reliability of the instrument, the DIGCOMP-PED test was evaluated using the Kuder–Richardson-21 indicator [33], which shows that the consistency of the answers obtained at the total level is acceptable (KR-21 = 0.60). Cronbach's alpha ($\alpha = 0.702$) indicates that the questionnaire has a good level of reliability. The degree of difficulty of the test is adequate (GD = 55.06%).

The sphericity of the relationships between the DIGCOMP items was analyzed through the Bartlett test, which was significant ($X2(210) = 696.305$, $p < 0.001$). In a complementary manner, the Kaiser–Meyer–Olkin sample adequacy index (KMO = 0.796) exceeded the critical value of 0.6. Taken together, these tests show that the indicators presented high correlations with each other, which allows us to factor the data collected.

For the exploratory factor analysis, the parallel analysis method was used [34], because the optimal number of factors to be extracted was estimated. Considering the international referential framework, we assumed that the possible factors to be extracted were correlated (i.e., areas of digital competence) and we used an oblique rotation (i.e., oblimin). The factorial analysis with the parallel analysis revealed that the competences presented, for the most part, low factorial loads (<0.40) and that the best was a unifactorial solution. After the factorial exploration analysis, we performed a confirmatory factorial analysis. Considering the ordinal nature of the competences, we used the WLSM (Weighted Least Squares) as a robust estimation method, which is more appropriate for data that do not distribute normally and are of an ordinal type. This analysis confirmed the recommendation of the parallel analysis and a unifactorial model was specified, in which, due to the interest of the investigation, we worked on the competences related to the safety area individually and not through statistical calculations applied to the area.

In the final instrument, the safety area incorporated 12 questions related to the four competences of the area, as seen in the diagram below (Figure 2).

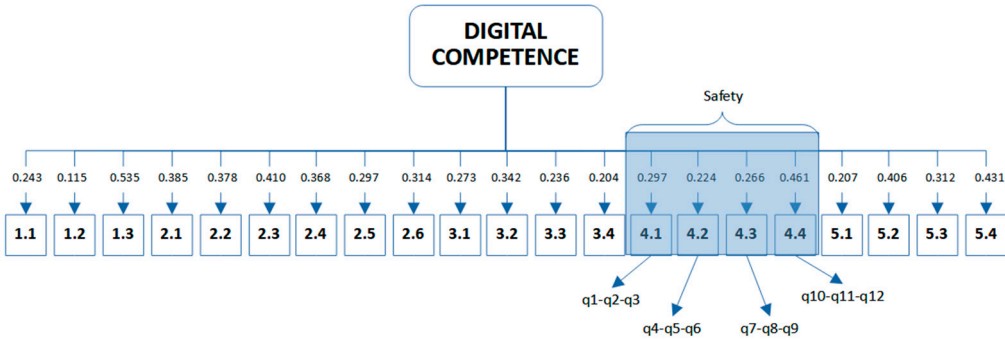

**Figure 2.** Digital competences, safety.

The questions were evaluated as correct or incorrect, with scores 0 and 1, respectively. Based on the above, the range of possible scores for each indicator was from 0 to 3, and for the safety area, it is from 0 to 12.

*2.2. Participants*

The sample of this study comprised 4360 higher education students (Table 1) attending undergraduate courses at the University of Santiago de Chile (USACH) in the years 2021 and 2022.

**Table 1.** Characterization of the sample.

|  | Total Sample (N = 4360) |  |
| --- | --- | --- |
| Variable | *n* | % |
| Admission |  |  |
| Female | 2021 | 56 |
| Male | 2022 | 44 |
| Gender |  |  |
| Female | 2.209 | 50.7 |
| Male | 2098 | 48.1 |
| Other/non-binary | 53 | 1.2 |
| Career |  |  |
| Medical Sciences | 427 | 9.8 |
| Biology and Chemistry | 216 | 5 |
| Law | 182 | 4.2 |
| Sciences | 263 | 6 |
| Engineering | 1430 | 32.8 |
| Administration and Economy | 693 | 15.9 |
| Humanities | 608 | 13.9 |
| Technology | 293 | 6.7 |
| Architecture | 92 | 2.1 |
| Basic university cycle (bachillerato) | 156 | 3.6 |
| Graduates' educational institution |  |  |
| Municipal | 1285 | 29.5 |
| Subsidized private | 2510 | 57.6 |
| Private paid | 482 | 11.1 |
| Delegated administration | 83 | 1.9 |
| Type of education |  |  |
| Vocational technical | 758 | 17.4 |
| Scientific humanist | 3547 | 81.4 |
| Other | 55 | 1.3 |

**Table 1.** *Cont.*

| | Total Sample (N = 4360) | |
|---|---|---|
| Years of access to technology | | |
|     Under 5 years | 506 | 11.6 |
|     6–10 years | 2512 | 57.6 |
|     11–15 years | 1121 | 25.7 |
|     Over 16 years | 221 | 5.1 |

*2.3. Procedure*

Participation in the research study was voluntary and the instrument was answered digitally, through the virtual platform used by the university. We applied the instrument to the newcomers during the virtual activities of immersion into university life that include courses in mathematics, language, and personal development, and one aimed at learning about the virtual campus and the resources that the university makes available to students. In this context, one of the activities is to answer the COMPDIG-PED survey.

*2.4. Data Analysis*

In SPSS (V 23.0), we conducted Student's *t*-test of independent samples to evaluate the mean differences in the scores obtained in the DIGCOMP competences related to safety area according to the year of admission. On the other hand, we used one-way ANOVA to evaluate the mean differences in the scores obtained in the DIGCOMP competences related to safety area according to gender, years of access to digital technologies, school institutions of origin, and type of formation in secondary education. We also applied Tukey's post hoc tests to determine for which pairs of variables the statistically significant differences detected by the ANOVA test in SPSS were located.

**3. Results**

In Table 2, at the level of safety competences, the competence of protecting personal data and privacy scored the highest on the 0–3 scale, with a mean of 2.58 and a standard deviation of 0.627. The lowest competences are health and well-being protection, with a mean of 2.10 and a standard deviation of 0.710, and protecting devices, with a mean of 2.10 and a standard deviation of 0.773.

**Table 2.** General results by the competence in the area of safety.

| Safety | Mean | Standard Deviation |
|---|---|---|
| Protecting device | 2.10 | 0.773 |
| Protecting personal data and privacy | 2.58 | 0.627 |
| Protecting health and well-being | 2.10 | 0.710 |
| Protecting the environment | 2.30 | 0.641 |

*3.1. Year of Admission to the University*

The level of achievement by year of university admission for each competence is presented in Table 3.

In the *t*-test of independent samples, we found that the protecting device competence presented significant differences according to the year of admission, where the score of 2021 (M = 2.12, DE = 0.774) was higher than that of 2022 (M = 2.07, SD = 0.771), $t$ (4358) = 1.98, $p < 0.005$. The protecting personal data and privacy competence presented significant differences, where the 2021 score (M = 2.54, DE = 0.653) was lower than that of 2022 (M = 2.63, SD = 0.589), $t$ (4276.5) = $-4.57$ $p < 0.001$. The protecting the environment competence showed significant differences, where the 2021 score (M = 2.24, DE = 0.662) was lower than that of 2022 (M = 2.36, SD = 0.607), $t$ (4358) = $-6.13$ $p < 0.001$.

**Table 3.** Mean and standard deviation by year of admission.

| | Admission 2021 | | Admission 2022 | |
| --- | --- | --- | --- | --- |
| | **Mean** | **SD** | **Mean** | **SD** |
| Protecting device | 2.12 | 0.774 | 2.07 | 0.771 |
| Protecting personal data and privacy | 2.54 | 0.653 | 2.63 | 0.589 |
| Protecting health and well-being | 2.09 | 0.722 | 2.11 | 0.695 |
| Protecting the environment | 2.24 | 0.662 | 2.36 | 0.607 |

*3.2. Gender*

The level of achievement by gender for each competence is shown in Table 4.

**Table 4.** Mean and standard deviation by gender.

| | Female | | Male | | Other/Non-Binary | |
| --- | --- | --- | --- | --- | --- | --- |
| | **Mean** | **SD** | **Mean** | **SD** | **Mean** | **SD** |
| Protecting device | 2.12 | 0.760 | 2.07 | 0.786 | 2.32 | 0.701 |
| Protecting personal data and privacy | 2.60 | 0.612 | 2.55 | 0.644 | 2.70 | 0.503 |
| Protecting health and well-being | 2.13 | 0.672 | 2.07 | 0.748 | 2.19 | 0.681 |
| Protecting the environment | 2.33 | 0.605 | 2.26 | 0.678 | 2.42 | 0.497 |

ANOVA analysis reveals that gender is related to the protecting device competence, $F_{(2, 4357)} = 4.58$, $p = 0.010$; protecting personal data and privacy competence, $F_{(2, 4357)} = 5.34$, $p = 0.005$; protecting health and well-being competence, $F_{(2, 4357)} = 4.35$, $p = 0.013$; and protecting the environment competence, $F_{(2, 4357)} = 6.72$, $p = 0.001$. In other words, there is a statistically significant relationship between all the competences of the safety area.

Tukey test post hoc analyses showed that women scored higher than men in protecting personal data and privacy ($p = 0.009$), 95% CI [01, 10]; in protecting health and well-being ($p < 0.014$), 95% CI [01, 11]; and in protecting the environment ($p = 0.002$), 95% CI [02, 11].

*3.3. Years of Access to Digital Technologies*

The level of achievement by the number of years of access to digital technologies for each competence is expressed in Table 5.

**Table 5.** Mean and standard deviation by years of experience using digital technologies.

| | Under 5 Years | | 6–10 Years | | 11–15 Years | | Over 16 | |
| --- | --- | --- | --- | --- | --- | --- | --- | --- |
| | **Mean** | **SD** | **Mean** | **SD** | **Mean** | **SD** | **Mean** | **SD** |
| Protecting devices | 2.06 | 0.842 | 2.12 | 0.768 | 2.08 | 0.755 | 2.09 | 0.751 |
| Protecting personal data and privacy | 2.47 | 0.692 | 2.59 | 0.621 | 2.58 | 0.619 | 2.65 | 0.548 |
| Protecting health and well-being | 2.04 | 0.754 | 2.11 | 0.703 | 2.09 | 0.713 | 2.21 | 0.662 |
| Protecting the environment | 2.19 | 0.735 | 2.31 | 0.626 | 2.30 | 6.30 | 2.35 | 0.611 |

The ANOVA analysis related the years of access to digital technologies with the protecting personal data and privacy, $F_{(3, 4356)} = 5.99$, $p < 0.001$; protecting health and well-being, $F_{(3, 4356)} = 3.10$, $p = 0.026$; and protecting the environment competences, $F_{(3, 4356)} = 5.96$, $p < 0.001$.

Tukey test post hoc analyses showed that people with less than five years of experience using digital technologies had lower scores in three digital competences of the safety area.

In the competence of protecting personal data and privacy, the group with less than 5 years of experience had lower scores than those with between 6 and 10 years of experience ($p = 0.001$), 95% CI [−19, −04]; than people with between 11 and 15 years of experience ($p = 0.011$), 95% CI [−19, −02]; and than people with more than 16 years of experience ($p = 0.003$), 95% CI [−31, −05].

For protecting health and well-being, the group with less than 5 years of experience had lower scores than those with more than 16 years of experience ($p = 0.017$), 95% CI [−32, −02].

For the competence of protecting the environment, the group with less than 5 years of experience had lower scores than the people with between 6 and 10 years of experience ($p < 0.001$), 95% CI [−21, −05]; than people with between 11 and 15 years of experience ($p = 0.007$), IC 95% [−20, −02]; and than people with more than 16 years of experience ($p = 0.010$), 95% CI [−29, −03].

### *3.4. School Institutions of Origin*

The level of achievement related to the educational institution of origin of the sample participants for each competence is expressed in Table 6.

**Table 6.** Mean and standard deviation by type of school institution according to its dependence/bond.

| | Municipal | | Subsidized Private | | Private Paid | | Delegated Administration | |
|---|---|---|---|---|---|---|---|---|
| | Mean | SD | Mean | SD | Mean | SD | Mean | SD |
| Protecting devices | 2.11 | 0.773 | 2.10 | 0.773 | 2.03 | 0.785 | 2.14 | 0.683 |
| Protecting personal data and privacy | 2.52 | 0.674 | 2.60 | 0.609 | 2.61 | 0.596 | 2.66 | 0.547 |
| Protecting health and well-being | 2.07 | 0.720 | 2.11 | 0.715 | 2.12 | 0.665 | 2.00 | 0.681 |
| Protecting the environment | 2.24 | 0.676 | 2.31 | 0.628 | 2.37 | 0.612 | 2.27 | 0.607 |

Based on the ANOVA analysis, the type of formation in secondary education is related to the protecting personal data and privacy, $F(3, 4356) = 5.36$, $p = 0.001$, and protecting the environment competences, $F(3, 4356) = 5.30$, $p = 0.001$.

Tukey test post hoc analyses showed that in the competence of data protection and privacy, people from municipal schools had lower scores than those from private subsidized ones ($p = 0.002$), 95% CI [−13, −02], and those that came from paid private establishments ($p < 0.050$), IC 95% [−17, 00].

For the protecting the environment competence, people from municipal schools had lower scores than those from private subsidized institutions ($p = 0.013$), 95% CI [−12, −01], and those who came from paid private schools ($p = 0.002$), 95% CI [−21, −03].

### *3.5. Type of Education*

The level of achievement related to the type of education of the participants in the sample for each competence is expressed in Table 7.

**Table 7.** Mean and standard deviation by type of institution according to its teaching modality.

| | Vocational Technical (VT) | | Scientific Humanist (SH) | | Other | |
|---|---|---|---|---|---|---|
| | Mean | SD | Mean | SD | Mean | SD |
| Protecting devices | 2.06 | 0.781 | 2.11 | 0.770 | 2.04 | 0.838 |
| Protecting personal data and privacy | 2.50 | 0.672 | 2.60 | 0.614 | 2.45 | 0.715 |
| Protecting health and well-being | 2.01 | 0.761 | 2.12 | 0.696 | 2.04 | 0.793 |
| Protecting the environment | 2.23 | 0.669 | 2.31 | 0.663 | 2.22 | 0.686 |

From the ANOVA analysis, the type of education in secondary education, from the perspective of the teaching modality, relates to the protecting personal data and privacy competence, $F(2, 4357) = 8.57$, $p < 0.001$; to the protecting health and well-being competence, $F(2, 4357) = 7.32$, $p = 0.001$; and to the protecting the environment competence, $F(2, 4357) = 6.26$, $p = 0.002$.

Tukey test post hoc analyses showed that the participants who had vocational technical formation (VT) scored lower than those from scientific humanist formation (CH) in the competence of protecting data and privacy ($p < 0.001$), 95% CI [−16, −04]; in the competence of protecting health and well-being ($p < 0.001$), 95% CI [−17, −04]; and in the competence of protecting the environment ($p = 0.002$), 95% CI [−15, −03].

## 4. Discussion

This study aimed to describe the level of digital competence in the safety area in first-year undergrads of a Chilean public university and its relationship with the year of admission to the university, gender, years of access to technologies, type of educational center from the point of view of dependence, and type of educational center from the point of view of the education provided.

The results suggest that the differences in the context variables are related to the competences related to safety in the students in the sample.

First, there is a trend for higher scores in the most recent entry, in 2022, than in the 2021 entry. In two competences, protecting personal data and privacy and protecting the environment, the 2022 students obtained higher scores with more marked differences than the 2021 students, who stood out in the first competence of protecting devices. Although comparing the two groups is insufficient for us to speak properly of a trend over time, we could infer that this difference is owing to the increasing massification of digital technology in the population, especially in the younger generations. In this sense, in line with Arachchilage [35], we understand that people must be aware and have conceptual and procedural knowledge to address the issues of computer safety.

Secondly, women scored higher than men in the four competences of the safety area. Three of them had statistically significant differences: protecting personal data and privacy, protecting health and well-being, and protecting the environment. These data are consistent with the results of the Ministry of Economy, Development, and Tourism of Chile [36] that working women use digital devices more than men. However, the higher use contrasts with women's current positions in technology companies and cybersecurity [37] in a study related to women's role in the technology sector—a special reference to technology companies, emerging jobs, and the field of cybersafety—which argues that the observed female leadership problems in technology companies are usually replicated and emphasized in the field of cybersecurity, in which women present little professional progress within organizations. Likewise, women's motivation to develop their professional careers in cybersafety declines due to the lack of support from bosses and colleagues. Findings from this study are related to previous research (e.g., Rebollo-Catalán, Mayor-Buzon, and García-Pérez' [25]). Results from this research suggest that women are more aware of and committed to the protection of their personal information, health, well-being, and environment in digital contexts.

On the other hand, the years of access to digital technologies are related to three competences of the safety area: protecting personal data and privacy, protecting health and well-being, and protecting the environment. The results showed that this experience of access and use only impacts people with less than five years of experience. They had lower scores than the other groups, which did not present significant differences. Based on these data, we infer that the basic digital skills linked to computer security are acquired, mainly, during the first stage in the use of the DTs, which also makes sense given that the differences are manifested in the three competences that require greater familiarization with this type of technology: Learning to protect the devices is the first thing that is needed to use DTs, and it is precisely in this competence that the groups do not differ significantly. The need to learn how to protect personal data and privacy appears when interacting with other people and using the internet for specific services that require a digital identity. The need for health protection appears when the potential damage that an inappropriate use of DTs can cause is discovered, just like the need to protect the environment appears when people realize the potential effects that the use and consumption of DTs have on the

environment. These results support the idea that it is important to guide an appropriate use of technologies, especially in those who still have little experience. Such a guide is essential because beginner users can develop an addiction when using digital resources. Technology addiction has similar manifestations to traditional addictions (Terán [11]). Moreover, video game addiction was included in the latest WHO [19] International Classification of Diseases (ICD-11). Additionally, the ESTUDES 2016–2017 survey conducted in Spain by the National Plan on Drugs revealed that compulsive internet use and online gaming have a negative impact on students [12].

On the other hand, Tucho and González [13] emphasize the environmental and socioeconomic impact of the manufacturing of technological devices, as well as the generation of electronic waste in poor regions of the planet. In this sense, the need to train citizens in digital skills, especially in computer security, is becoming increasingly evident, considering the various impacts that ICTs have on society.

Finally, according to the type of educational institution from the point of view of its dependence and the type of education it delivers, we find that, precisely, the schools that receive students from the most vulnerable sectors present lower results in digital competences related to computer security.

Students from municipal schools in Chile, whose average vulnerability index between 2016 and 2022 is the highest in the system, according to the Measurement of Multidimensional Student Vulnerability published by the Junta Nacional de Auxilio Escolar y Becas [38], obtained lower scores, both in protecting personal data and privacy and protecting the environment.

From the perspective of the type of education provided, students from vocational technical schools, which also serve vulnerable sectors, obtained the lowest scores in protecting personal data and privacy, protecting health and well-being, and protecting the environment.

These results agree with the deep link that, according to [39], exists in Chile between the educational system and social classes and is a systemic reflection of the social structure through education. The educational experience as a structuring of life experiences and generational trajectories in the Chilean model has contributed to social segments with certain cultural identities and educational performance, typical of their social group of origin [40].

## 5. Conclusions

Digital competences linked to computer safety are crucial and strategic for the process of technological transformation that we are experiencing. They constitute a basis for the organic and responsible incorporation of DTs in society.

This work helped us observe that the first-year university students attending several undergraduate courses at a Chilean public university undergo a series of context variables that are related to the development of the competences, among which we list gender, where women have higher averages than men; the years of access to technology, with better results for people with more years of experience; and the type of school institution, highlighting whether they are private paid or subsidized private schools and, according to their teaching modality, the scientific humanist school institutions.

The emphasis on specific variables can prove beneficial for the development of competences and the enhancement of educational institutions, highlighting the elements that require more intensive and urgent attention.

This article provides an empirical analysis of the development of digital competences in Chilean university students, emphasizing the area of computer security. By adopting COMPDIG-PED, the study allows us to comprehend how contextual factors, such as gender, previous experience with technology, and type of educational institution influence these competences. The findings highlight the relevance of digital competence training, both in education and in the transition to a more digitalized society. Furthermore, they can

contribute to the improvement of future formative programs and interventions related to the development of the students' digital competence.

Despite its contributions, this study has some limitations. The sample comprised only first-year students from a Chilean public university. Thus, the findings may not be generalizable to other contexts or educational levels. Also, although some contextual variables were examined, there are others that were not considered in this study that could influence digital competences, such as the quality of digital technology education that students received in their secondary education.

The results obtained in this research open several lines for future work. It would be valuable to conduct longitudinal studies to assess how digital competences evolve over time and how they respond to various educational interventions. It would also be beneficial to expand this research to other universities and educational levels to better understand the generalizability of the findings. In addition, it would be interesting to investigate the influence of other contextual variables on digital competences, as well as to deeply explore the relationship between digital technology education received at the secondary level and digital competences in higher education.

**Author Contributions:** Methodology, J.S.Q.; Formal analysis, M.R.P.; Investigation, M.R.P.; Resources, J.S.Q.; Writing—review and editing, C.C.M. All authors have read and agreed to the published version of the manuscript.

**Funding:** This research received no external funding and the APC was funded by the authors.

**Institutional Review Board Statement:** The study was conducted in accordance with the Declaration of Helsinki, and approved by the Comité de ética institucional Universidad de Santiago de Chile (Resolución N°011494).

**Informed Consent Statement:** Informed consent was obtained from all subjects involved in the study.

**Data Availability Statement:** The data that support the findings of this study are available from the corresponding author upon request.

**Conflicts of Interest:** The authors declare no conflict of interest.

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
