# Peer review of "Development of Digital Competences in Students of a Public State-Owned Chilean University Considering the Safety Area"

_education, doi:10.3390/educsci13070710_

Round 1
Reviewer 1 Report
As the study uses DigComp 2.1 as the reference, why doesn’t it use the framework’s language, competence, instead of indicator (a word that doesn’t show up on the framework)?
The article also uses a variety of words and expressions to describe the competence area, which turns the reading confusing even for senior digital competence researchers: security, safety, network security, network safety, information security, computer security indicators (title), information security indicators (abstract)
This said, “Computer Security Indicators” does not help to make the title clear.
This study was conducted in a Chilean public university, the second biggest state university in the country, to describe the information security indicators of digital competence and its relationship with the year of admission to the university of students, observing gender, years of access to digital technologies, and the type of educational center of origin, from the point of view of dependency and education modality. – this is a long sentence involving many variables to start an abstract; concise sentences would be easier to read.
Isn’t competencies the plural of competency? But the article uses competence.
Why two paragraphs for the abstract?
it is essential to educate people to use digital devices and tools and take protective measures and strategies necessary for this interconnected world, which goes beyond protecting passwords or identity in social networks; it includes aspects such as health care or environmental sustainability – this (as the article itself) helps to widen the view for the DigComp competence area security
knowledge, skills, attitudes, strategies, and values – DigComp included the first 3 ones; it is important to explain where strategies and values come from and discuss it a little bit
Currently, various instruments can assess DC in university students, such as INCOTIC 2.0, created in Spain and designed to perform a self-diagnostic assessment of DC in first-year undergrads [26]. In Latin America, an adapted version is used: INCOTIC-LA, which is in the pilot stage, being applied in universities on the continent. Another instrument is ACUTIC, developed to study university students' attitudes toward ICTs [27] — isn’t it possible to include links to these instruments, if available, besides the references?
Most instruments applied to describe digital competencies are of self-perception, in which students have the highest perceptible level. For this reason, COMPDIG-EDUSUP was designed as a standard assessment instrument, which, unlike an appreciation scale, assesses the responses to specific situations of DC deployment in dichotomous parameters (correct and incorrect). – a critique of the self-perception instruments would highlight the importance of the proposed instruments and the article itself
What is the difference between COMPDIG-EDUSUP and COMPDIG-PED?
we used DIGCOMP 2.1 – why not 2.2? As it was published in 2022, there is a need to justify the option.
Why the choice for DigComp if it is not a framework that focuses on students? Although it is usually employed for assessing students' DC, there is a need to explain the choice.
Congratulations on the design, development, and careful validation of the instrument and the description of these processes.
63 items – how long does it take for students to answer the whole instrument? Aren’t there too many questions?
Is there a need to repeat in the text the information provided in Table 1? Couldn’t the text supplement and/or analyze data instead of only repeating numbers?
Is there a need for 3.1 if there is only this section?
Secondly, contrary to what one might think, women scored higher than men in the four information safety indicators. – where does this “one might think” comes from? I suggest rephrasing this, as it can be a door for criticism.
The discussion related to [36] is not clear. There is a need to clarify and deepen it.
The discussion introduces only four references [35] [36] [37] [38] to support its claims. It is too little compared to the richness of the whole article. Also, because of that, but not only, the discussion is too poor. In some cases, it involves repetition of the analysis. Besides, the Discussion flows item by item without even connecting the results of the analysis.
Measurement of Multi-dimensional Student Vulnerability published by the Junta Nacional de Auxilio Escolar y Becas (2023), obtained lower scores, both in Protecting personal data and privacy and Protecting the Environment. – where is the reference for this?
Conclusions. As per the Discussion, too poor. It lacks highlighting the article's contributions (including the design, development, and validation of the instrument), limitations of the research, and future works.
Comments on vocabulary made on the previous section.
Reviewer 2 Report
The article is original research manuscripts
The title of paper has 12 words. The Abstract has 176 words
The paper is following chapter:
1. Introduction
The introduction chapter has briefly placed the study in a broad context and highlight why it is important.
The Introduction chapter should define the purpose of the work and its significance, including specific hypothesis being tested.
Digital competencies, security area
2. Materials and Methods
Specific objectives:
Instrument to assess digital competence in pedagogy students
Participants
Procedure
Data analysis
3. Results
3.1. Descriptive statistics associated with the achievement of safety-related indicators
3.1.1 Year of admission to the university
3.1.2 Gender
3.1.3 Years of access to digital technologies
3.1.4 School institutions of origin
3.1.5 Type of education
4. Discussion - ok
5. Conclusions - ok
References
There is no consistency in the notation of chapters and subchapters and the formatting of their titles
The Introduction chapter should define the purpose of the work and its significance, including specific hypothesis being tested.
The bibliography is not in alphabetical order
